# Lipocalin-2 in Triple-Negative Breast Cancer: A Review of Its Pathophysiological Role in the Metastatic Cascade

**DOI:** 10.3390/ijms262210938

**Published:** 2025-11-12

**Authors:** Diandra T. Keller, Ralf Weiskirchen, Sarah K. Schröder-Lange

**Affiliations:** Institute for Molecular Pathobiochemistry, Experimental Gene Therapy and Clinical Chemistry (IFMPEGKC), RWTH University Hospital Aachen, D-52074 Aachen, Germany; dikeller@ukaachen.de

**Keywords:** LCN2, breast cancer, TNBC, metastasis, organ-tropism, signaling

## Abstract

Lipocalin-2 (LCN2) is a 25 kDa glycoprotein that has been shown to be a multifunctional player in the metastasis of triple-negative breast cancer (TNBC). In physiological contexts, LCN2 exhibits bacteriostatic properties and plays key roles in iron homeostasis and the transport of hydrophobic molecules. However, several studies have shown that aberrant LCN2 expression is associated with poor prognosis in various malignancies, including breast cancer, which is the most common cancer in women worldwide and can be classified into four molecular subtypes. Among these, TNBC represents a disproportionately aggressive subtype characterized by poor prognosis and high metastatic potential. Although LCN2 has been extensively studied in breast cancer overall, its specific role in TNBC progression and metastasis is only beginning to be understood. Recent evidence suggests that LCN2 contributes to several tumor-promoting processes such as angiogenesis, therapy resistance and modulation of the tumor microenvironment. Moreover, LCN2 appears to influence organ-specific metastasis, particularly to the lung and brain, while its role in liver and bone dissemination remains unclear. Collectively, current data identify LCN2 as a critical mediator of TNBC progression and highlight its potential as a prognostic factor and modulator of disease progression. This review aims to summarize insights from both in vitro and in vivo studies, with particular focus on the role of LCN2 in the metastatic cascade, while also addressing existing research gaps and critically evaluating the current findings.

## 1. Introduction

Lipocalin-2 (LCN2) is a member of the lipocalin superfamily, a group of circulating proteins [1]. It was first isolated from human neutrophil granules at sites of infection and inflammation, hence the name neutrophil gelatinase-associated protein (NGAL) was given [2].

LCN2 exists in several molecular forms: a 25 kDa monomer, a 46 kDa disulfide-linked homodimer, and a 135 kDa disulfide-linked heterodimer with neutrophil gelatinase B, also known as matrix metalloproteinase 9 (MMP-9) [2,3]. It is important to note that only human LCN2 can form dimers, because murine LCN2 lacks Cys^87^, which is necessary for dimer formation [4,5]. Translation to the clinic must account for structural differences between human and murine LCN2, particularly the absence of Cys^87^ in mice, preventing dimer formation, which may limit the preclinical relevance. Humanized models or advanced in vitro systems are needed to fully capture LCN2’s metastatic role in triple-negative breast cancer (TNBC).

### 1.1. Physiological Functions of LCN2

LCN2 has been shown to exert several physiological effects. Notably, it displays bacteriostatic properties: Goetz et al. demonstrated that LCN2 binds bacterial catecholate-type iron siderophores (e.g., enterochelin), contributing to the innate immune system’s iron depletion approach [6]. Enterochelin, secreted by bacteria to trap iron, is competitively bound by LCN2, which delivers the enterochelin/Fe^3+^ complexes to host cells for degradation [7].

Furthermore, LCN2 can bind to mammalian siderophores and act as an iron carrier that helps maintain intra- and extracellular iron homeostasis, providing transferrin-independent iron transport [8]. This ability allows LCN2 to deliver iron into cells, which can prevent cellular damage and promote proliferation and survival under physiological conditions [9].

Structurally, members of the lipocalin family are characterized by their ability to bind and transport small hydrophobic molecules, including fatty acids, steroids, hormones, and retinoids [1]. These ligands bind to the characteristic lipocalin fold, an eight-stranded antiparallel β-barrel that forms a hydrophobic pocket. The hydrophobic ligands are released upon binding of LCN2 to its cellular surface receptors [10,11].

To date, six putative LCN2 receptors are described that differ in their structure and chromosomal location: neutrophil gelatinase-associated lipocalin receptor (NGALR, SLC22A17; 14q11.2), low density lipoprotein-related protein 2 (LRP2; 2q31.1), LRP6 (12p13.2), melanocortin 1 receptor (MC1R; 16q24.3), MC3R (20q13.2) and MC4R (18q21.32) [12,13]. While the functional role of LCN2 is well described, the precise downstream signaling pathways upon receptor binding remain largely unknown, as reviewed by Schröder et al. [12].

LCN2 is expressed throughout various organ systems, such as the respiratory system, including the nasopharynx and bronchus [14], the gastrointestinal tract [15], kidney, and bladder [16], male reproductive tissues (testes) [17] and female reproductive tissues (particularly the cervix) [18], and bone marrow and lymphoid tissues [15].

### 1.2. Pathological Functions of LCN2

LCN2 expression is significantly increased in several neoplastic epithelial tissues, such as the breast [19], pancreas [20], colon [21] and ovary [22]. In particular, aggressive subtypes of breast cancer (BC) are associated with high LCN2 expression, highlighting it as a key regulator of metastasis formation [23]. Cancer cells exploit the iron-transporting capability of LCN2 by inducing macrophages to secrete high levels of LCN2-bound iron, a mechanism linked to tumor formation, metastasis and recurrence in BC patients [24].

As mentioned earlier, LCN2 forms heterodimers with MMP-9 [3]. MMP-9 possesses gelatinase activity, allowing it to degrade type IV collagen in the basement membrane. This process is a crucial step in tumor invasion and subsequent metastasis [25]. A study by Bahrun et al. showed that high levels of LCN2 in BC patients correlate with elevated levels of MMP-9 [26]. Similarly, high MMP-9 expression has been associated with advanced tumor stage and lymph node metastasis in BC [25].

### 1.3. Breast Cancer

BC is the most commonly diagnosed cancer in women worldwide. According to GLOBOCAN 2020 data, BC accounted for approximately 2.3 million new cases globally, representing 11.7% of all cancer cases and surpassing lung cancer as the most frequent malignancy overall [27]. Among women, it remains the leading cancer type both in incidence and mortality, responsible for almost 685,000 deaths in 2020. In Germany alone, 74,016 new cases and 20,601 deaths were reported in 2022, highlighting its significant national disease burden [28]. BC is a heterogeneous neoplasm with multiple subtypes commonly characterized by the presence or absence of the hormone receptors estrogen (ER), progesterone (PR) and human epidermal growth factor (HER2). Accordingly, the subtypes luminal A (ER^+^/PR^+/−^/HER2^+^, Ki67 < 14%), luminal B (ER^+^/PR^+/−^/HER2^+^, Ki67 > 14%), HER2-positive (ER^−^/PR/HER2^+^), and triple-negative (ER^−^/PR^−^/HER2^−^) are widely recognized [29,30]. Although the role of LCN2 in BC has been extensively studied, its specific function in the metastatic progression of TNBC remains poorly defined [19,31,32]. Due to its aggressive behavior, high tendency to spread and limited treatment options, gaining insight into the molecular mechanisms that drive TNBC progression is crucial for improving clinical outcomes.

This review aims to summarize current insights into TNBC metastasis in both in vitro and in vivo models, with a particular focus on the emerging but understudied role of LCN2 in this context.

## 2. Subtypes of Triple-Negative Breast Cancer

Although TNBC accounts for only 10–17% of all breast cancers, it represents a disproportionately aggressive and clinically challenging subtype [33]. TNBC is characterized by the absence of ER, PR and HER2 expression, making it unresponsive to hormone therapies or HER2-targeted treatments. This limits therapeutic options primarily to chemotherapy, which is often associated with poor response rates and early recurrence [34]. Furthermore, TNBC exhibits a high histological grade [35], increased proliferative activity [36] and a greater tendency for visceral and brain metastases [37,38]. Its molecular heterogeneity and tendency to affect patients younger than 40 years and certain ethnic groups underscore the need for a deeper understanding and the development of targeted therapeutic strategies [38,39].

Given the clinical complexity and therapeutic challenges associated with TNBC, efforts have been made to further stratify this heterogeneous disease based on molecular characteristics: Cluster analysis of tumor samples from 587 TNBC patients performed by Lehmann et al. classified TNBC into six subtypes: basal-like 1 (BL-1), basal-like 2 (BL-2), mesenchymal (M), mesenchymal stem-like (MSL), immunomodulatory (IM), and luminal androgen receptor (LAR) [40]. Later, they further refined these assumptions and grouped MSL and IM under the category for the mesenchymal subtype M, since the transcripts of these subtypes are derived from tumor-infiltrating lymphocytes and tumor-associated stromal cells [41]. They also categorized TNBC cell line models as representatives of these subtypes, thereby providing an accurate cell model for clinical treatment [40]. This study by Lehmann et al. highlights the heterogeneity of TNBC and provides useful data for targeted therapy of different TNBC patients (Table 1).

## 3. Metastasis of Triple-Negative Breast Cancer

The versatile functions of LCN2 have attracted the attention of several researchers, who have suggested a specific role for LCN2 in TNBC. As early as 2009, Yang and colleagues linked elevated LCN2 levels in tissue and urine samples from BC patients to increased invasiveness [19].

TNBC has a tendency to metastasize to distant organs, referring to metastases outside the ipsilateral breast, chest wall and regional lymph nodes [54]. Between 6 and 10% of patients have metastases at the time of their primary diagnosis, while 30% will relapse or metastasize [55]. Despite active treatment, 25% still experience recurrence and the development of distant metastases [56]. The 5-year overall survival rate in metastatic TNBC is 4–20% in Western countries [57,58].

### 3.1. Molecular Mechanisms

#### 3.1.1. The Metastatic Cascade

Metastasis is generally divided into five sequential steps [59]: (1) Invasion of cancer cells from the initial tumor is the first step in the metastatic cascade. In this context, LCN2 interacts with MMP-9, stabilizing it and effectively exerting higher gelatinolytic activity on the extracellular matrix, enhancing matrix degradation, tumor progression and metastasis in BC [60]. (2) During intravasation, vascular endothelial growth factor (VEGF) secreted by cancer cells increases the permeability of tumor blood vessels, allowing cancer cells to enter the bloodstream [61]. Yang and coworkers linked LCN2 expression in TNBC cells to the induction of VEGF, promoting neovascularization and vascular permeability, thus playing an important role in cancer metastasis [62,63]. Furthermore, researchers have shown that breast cancer progression is directly linked to increased angiogenesis [64,65]. Angiogenesis is the formation of new blood vessels, which is critical for malignant cells to improve nutrient supply and gas exchange [66]. As previously mentioned, overexpression of LCN2 in MDA-MB-231 cells led to an upregulation of VEGF expression [62]. To elucidate the underlying regulatory mechanism, the role of hypoxia-inducible factor 1 alpha (HIF1α) was investigated. LCN2 knockdown resulted in reduced HIF1α protein levels, while silencing of HIF1α correspondingly decreased VEGF expression. These findings suggest that LCN2 promotes VEGF expression, at least in part, through the upregulation of HIF1α, positioning HIF1α as a downstream effector in the LCN2-mediated regulation of angiogenic signaling. (3) In the blood or lymphatic circulation, cancer cells acquire resistance to anoikis, a form of apoptosis triggered by the absence of survival signals [67]. Survival signals are, among other things, generated by the interaction of cell adhesion with the extracellular matrix [63]. Cancer cells adhere in the microvasculature when their adhesive capacity is greater than the shear forces of the blood flow [68]. In order to colonize tissue, cancer cells adhere to the luminal surface of a capillary via a specific set of adhesion molecules [69]. (4) This step, known as extravasation, involves the migration of cancer cells across the endothelium [70]. (5) Finally, malignant cells colonize by remodeling the tissue at the site of metastasis, known as metastatic niche formation [71]. A general strategy of cancer cells is metabolic reprogramming, through which they obtain nutrients from their environment [72]. It is well established that sites of future metastasis are selectively and actively modified by the primary tumor before metastatic spread. The pre-metastatic niche is primed by factors secreted by the primary tumor to facilitate the survival of seeding cancer cells [73]. The complex process of metabolic priming of the pre-metastatic niche is poorly understood. However, TNBC cells secrete vesicles carrying miR-122 microRNA that suppress glucose uptake of niche cells in vitro and in vivo. miR-122 suppresses the expression of pyruvate kinase and the glucose transporter *GLUT1,* leading to an increase and accumulation of glucose in the tissue for the cancer cell [74]. These findings highlight a critical mechanism by which TNBC cells actively prepare distant tissues for metastatic colonization through metabolic priming. Understanding this form of metabolic reprogramming is essential for developing targeted therapies aimed at disrupting the pre-metastatic niche and limiting metastatic progression.

Roshanzamir et al. reported that metastatic TNBC cells have a metabolic network that is an intermediate state between the metabolic network of the tissue of origin and the distant metastatic site [75]. Furthermore, they demonstrated that TNBC cells that metastasize to the liver have enriched gene sets associated with oxidative phosphorylation, fatty acid metabolism, bile acid metabolism, and xenobiotic metabolism. Cells metastasizing to the lung shared some of these enriched gene sets but had higher expression of genes related to the complement system and KRAS (Kirsten rat sarcoma virus) signaling. Interestingly, primary breast cancer cells were more enriched in genes associated with proliferation, cell cycle control, epithelial to mesenchymal transition (EMT), and tumor necrosis factor alpha (TNF-α) or interferon signaling pathways [75]. These findings highlight the metabolic plasticity of TNBC cells, enabling them to adapt their metabolic programs to their niches.

LCN2 is implicated in metabolic reprogramming in various cancers, although its role in TNBC remains unclear. In colorectal cancer, metastatic cells exhibit reduced LCN2 expression, and LCN2 knockdown enhances glucose uptake and lactate production while downregulating energy metabolism genes [76]. Conversely, TNBC shows upregulation of genes related to glucose uptake and lactate metabolism [77]. The association between LCN2 and metabolic reprogramming in TNBC remains largely unexplored, with its effects appearing to be tumor- and tissue-specific. In TNBC, where glycolytic gene expression is already elevated, the specific role of LCN2 in niche priming is still unknown, underscoring a significant gap in current research.

In summary, the data suggest that LCN2 contributes to the metastatic cascade, as schematically illustrated in Figure 1. However, the underlying mechanisms through which LCN2 modulates cellular signaling remain largely undefined.

#### 3.1.2. Interaction with Tumor Microenvironment

The tumor microenvironment (TME) comprises a complex network of various cell types, extracellular matrix, blood vessels and soluble factors that surround and interact with the tumor [78]. Cancer cells modulate stromal and immune cells within the TME to promote growth, immune evasion, and angiogenesis, ultimately supporting tumor progression and metastasis.

Migration cues for tumor cell invasion through the stroma are provided by various factors, including extracellular matrix (ECM) proteins [79]. Published studies have focused on a single ECM cue, leaving the complex interplay between different ECM proteins and tumor cells largely unexplored [80,81,82]. Baskaran et al. set out to unravel the process of invasion and metastasis mediated by ECM protein signaling [83]. The researchers investigated the crosstalk between cancer cells and key ECM proteins, including collagen I, collagen IV, fibronectin, and tenascin C. These proteins have previously been identified as highly expressed in breast tumors [79,84]. By defining specific morphological parameters, they demonstrated that collagen I, fibronectin, and collagen IV increased cell area and promoted elongation, which may be associated with the invasive potential of the cell lines. Furthermore, the study identified the morphological features of different BC cell lines that determine 3D invasion in response to ECM proteins, thereby advancing our understanding of tumor-ECM interactions in metastasis formation [83].

The TME is a complex ecosystem in which cancer cells form the core surrounded by a variety of non-malignant cells, such as immune cells [85,86,87], cancer-associated fibroblasts [88], vascular and lymphatic endothelial cells [89,90], but also other tissue-specialized cell types, such as neurons [91]. Secretion of cytokines and angiogenic factors in cross-talk between cancer cells and stromal cells plays an important role in TNBC promotion and metastasis [92,93]. In the search for novel therapeutic targets, Malone et al. investigated the secretome of TNBC cells co-cultured with lymphatic endothelial cells (LEC), microvascular endothelial cells (MEC), fibroblasts, and macrophages [94]. LCN2 was one of the top 5 factors secreted by LEC, MEC, and fibroblasts when cross-talking with TNBC cells. LCN2 might be a key factor in TNBC progression and metastasis, as the inhibition of LCN2 by an anti-LCN2 antibody resulted in reduced cell growth and migration.

As previously mentioned, the TME comprises various cell types, including innate immune cells like macrophages and neutrophils, as well as cells from the adaptive immune system, such as T and B cells [95,96]. In addition to its bacteriostatic effects, recent research indicates that LCN2 also plays a role in both the innate and adaptive immune responses. Tumor-associated macrophages (TAMs) are essential for maintaining iron homeostasis with the M2-phenotype in particular promoting tumor growth and resistance to therapy by releasing high levels of LCN2 in the TME, providing iron to cancer cells [97,98]. Moreover, LCN2 secreted by TAMs stimulates LEC proliferation, lymphatic vessel expansion, and increased metastasis of BC [99]. These findings underscore the significance of LCN2 in regulating iron levels and promoting lymphangiogenesis.

In normal physiology, LCN2 is an innate immune protein secreted by neutrophils and macrophages in response to inflammation and infection [100]. However, cancer cells hijack the innate immune functions of LCN2 within the TME. Neutrophil plasticity allows these cells to adjust their phenotype and function according to the surrounding TME, resulting in a spectrum of effects that can either support (N2-phenotype) or inhibit (N1-phenotype) tumor progression [101,102]. A study by Wei et al. positively correlated neutrophil count with LCN2 expression levels in BC patients [103]. Higher neutrophil counts were associated with increased LCN2 expression, which increased with advancing tumor stage. Tumor cells secrete cytokines (e.g., IL-8, granulocyte colony-stimulating factor (G-CSF)) to recruit and activate neutrophils [104]. Activated neutrophils release LCN2, which further enhances tumor aggressiveness and promotes a pro-tumor inflammatory milieu [101]. This creates a positive feedback loop between neutrophils and LCN2, driving TNBC progression and potentially correlating with advanced disease stages. Neutrophils also support the recruitment and activation of regulatory T cells (T_regs_) and myeloid-derived suppressor cells (MDSCs), further suppressing anti-tumor immunity [102,105]. Thus, LCN2 serves as a key link between innate immune activation and cancer progression, highlighting how innate immune components can be reprogrammed by tumors to favor their own survival and spread.

Although LCN2 primarily acts as an effector molecule of the innate immune system, it indirectly modulates the adaptive immune response through its influence on cytokine signaling and iron homeostasis. Research has largely focused on the tumor cell–intrinsic functions of LCN2, while its roles in cells of the adaptive immune system have received considerably less attention. Che and coworkers discovered that anti-tumor immunity is suppressed by the differential expression of iron transporters [106]. They showed that LCN2 expression is upregulated in tumor-infiltrating CD4^+^ T cells. Furthermore, increased LCN2 secretion by T cells into the TME leads to decreased intracellular iron and, consequently, to apoptosis in T cells. Cancer cells deploy this mechanism by capturing LCN2-bound iron, leading to increased tumor proliferation. Based on these findings, LCN2 is considered a potential immunotherapeutic target. However, research is not explicitly focused on TNBC, but rather on mechanisms in other entities such as colorectal cancer. Further investigations are needed to determine the extent to which these mechanisms can be transferred to other types of tumors.

Another study by Floderer et al. suggested a dual role of dendritic cell-secreted (DC) LCN2 in T-cell immunity: it promotes apoptosis in CD8^+^ T cells, yet is essential for their priming and for driving a TH1-skewed environment [107]. The authors proposed that LCN2 acts as a modulator of the balance between immune stimulation and suppression, influencing how adaptive immunity is shaped following DC activation. However, most experiments were conducted in vitro using transgenic T-cell models, which may not fully reflect in vivo immune dynamics.

Overall, LCN2 plays a complex and context-dependent role in the adaptive immune system, modulating T-cell survival, apoptosis, and polarization through its effects on iron homeostasis and cytokine signaling. Nevertheless, research on LCN2 across both innate and adaptive immunity remains limited, and its roles in other immune cell types are still poorly understood. This highlights the need for further investigation to fully elucidate its impact on the immune system.

### 3.2. Phenotypic Plasticity

The expression of mesenchymal markers has been reported by several studies to represent EMT in TNBC [108,109,110]. Conversely, the formation of metastases to distant organs is thought to require the transition from a mesenchymal to an epithelial phenotype (MET) through the downregulation of mesenchymal markers [111,112].

Grasset et al. demonstrated that TNBC cells rely on the mesenchymal marker vimentin for invasion in 3D in vitro cultures, a finding corroborated by enhanced metastasis formation in vivo [113]. Notably, vimentin silencing reduced primary tumor growth, while its sustained expression appeared to inhibit metastasis at later stages. These findings suggest that upregulation of this marker during EMT may facilitate early invasive behavior, whereas its downregulation could be beneficial during the establishment of distant metastases. Despite these dynamic changes, TNBC tumors and their metastases exhibited a heterogeneous mix of epithelial and mesenchymal cell states, rather than a complete reversion to a purely epithelial or mesenchymal phenotype [113]. This phenotypic diversity in metastatic TNBC may underlie its notable resistance to treatment.

MDA-MB-231 cells, a commonly used TNBC model, exhibit a mesenchymal phenotype characterized by high expression of mesenchymal markers like vimentin and absence of epithelial markers such as E-cadherin [114]. Silencing of LCN2 in these cells leads to decreased vimentin levels and a shift toward a more compact, epithelial-like morphology, indicative of MET, accompanied by reduced migratory capacity [19,115]. In contrast, MCF-7 cells that represent the luminal A subtype (ER^+^/PR^+^/HER2^−^), display lower LCN2 expression, retain epithelial morphology, and express high levels of E-cadherin, though they also express vimentin to a lesser extent [116]. These findings suggest that LCN2 may play a critical role in maintaining the mesenchymal and invasive phenotype of TNBC cells, thereby contributing to their aggressive behavior and metastatic potential. However, it remains unclear whether it is less important that LCN2 levels are inherently high or low, and more relevant how these basal expression levels change during interactions with the TME or other cancer cells. It is important to clarify the dynamic processes of EMT and MET, which involve reversible transitions between epithelial and mesenchymal states. LCN2 primarily supports the maintenance of the mesenchymal phenotype in TNBC, promoting the expression of mesenchymal markers like vimentin and enhancing migratory and invasive behavior. While it stabilizes this cell state, LCN2 does not appear to directly drive the reversible EMT or MET. Its effects may also be modulated by the TME, indirectly influencing cell plasticity and metastatic potential. In line with this, a study by Yang et al. demonstrated that silencing LCN2 in MDA-MB-231 cells led to reduced cell migration and a shift toward a more epithelial-like morphology [19]. The authors suggested that LCN2 contributes to the mesenchymal phenotype and invasive behavior of MDA-MB-231 cells. However, the molecular mechanism of the relationship between LCN2 and enhanced migration and invasion through increased vimentin expression is still limited.

A recently published study investigated how metabolic stress influences phenotypic plasticity in TNBC [117]. When deprived of glutamine (Gln), Gln synthetase (GS) was up-regulated, leading to an epithelial–mesenchymal-like state and resistance to ferroptosis. GS synthesizes Gln from glutamate and ammonia, which is crucial for cancer cell proliferation [118]. The authors of this article identified the ATF4-NUPR1-LCN2 axis (ATF4: activating transcription factor 4; NUPR1: nuclear protein 1) as a key downstream pathway. Upregulation of GS and LCN2, along with downregulation of transferrin receptor (TFRC), lowers intracellular iron levels, inhibiting ferroptosis. Notably, knockdown of *LCN2* reversed the GS-induced plasticity and metastatic potential, emphasizing the central role of LCN2 in the adaptation process. Although the study offers valuable mechanistic insights, it primarily focuses on short-term nutrient stress and mouse models, leaving its clinical relevance in human metastasis to be further validated.

## 4. LCN2 as a Prognostic and Therapeutic Target in Triple-Negative Breast Cancer

### 4.1. Chemoresistance in TNBC

30–50% of TNBC patients develop resistance to neoadjuvant chemotherapy, leading to poor overall survival [119,120]. To address this challenge, novel therapeutic strategies are being explored. One such approach by Sun and colleagues involved the chemical modification of cyclin-dependent kinase inhibitors. Their compound G-4, a derivative of roscovitine, showed high sensitivity to TNBC cells [121]. Treatment with G-4 significantly decreased cell viability, proliferation, and migration in both in vitro and in vivo studies. Importantly, the expression of LCN2 was reduced in a concentration-dependent manner. Knocking out LCN2 in MDA-MB-231 cells using siRNA increased the IC_50_ of G-4 by 100-fold, while overexpression of LCN2 decreased the value by about two-fold. Since lipid reactive oxygen species (ROS) and ferroptosis-related cell death were absent in LCN2-deficient cells, the authors suggested that G-4 induces ferroptosis through LCN2 downregulation (Figure 2A) [121]. These findings indicate that targeting LCN2 or its regulatory pathways could be a promising strategy to overcome chemoresistance and enhance the sensitivity of TNBC cells to ferroptosis-inducing agents.

Another study conducted by our group showed that LCN2 expression is downregulated in doxorubicin-resistant 4T1 TNBC cells [122]. This downregulation was accompanied by increased activation of bone morphogenetic protein (BMP) signaling. Specifically, BMP2, a known pro-tumorigenic ligand in BC, was found to suppress LCN2 expression in resistant cells. These results suggest that LCN2 downregulation is a characteristic of acquired chemoresistance in this TNBC model (Figure 2B). However, the re-induction of LCN2 expression in response to inflammatory stimuli like IL-1β indicated that its regulation is not permanently lost but remains responsive to microenvironmental cues. This highlights the therapeutic potential of targeting LCN2 or its upstream inflammatory signaling pathways to overcome chemoresistance in TNBC.

### 4.2. Epigenetic Regulation of LCN2

Hypermethylated in cancer 1 (HIC1) is a tumor suppressor gene that is only silenced in TNBC compared with other molecular subtypes [123,126]. HIC1 is regulated upstream by p53 through a regulatory feedback loop that deacetylates and inactivates p53 [127]. Downstream targets of HIC1 include genes responsible for developmental and cell cycle control, but its regulatory mechanism in different breast cancer subtypes remains unclear [128].

Cheng et al. showed that LCN2 is a downstream target of HIC1 [123]. The in vitro results showed that HIC1 is recruited to the LCN2 promoter to repress its expression (Figure 2C). Lentiviral induction of HIC1 in MDA-MB-231 cells resulted in reduced invasiveness in vitro and reduced metastatic potential in vivo. Overexpression of LCN2 in MDA-MB-231 cells increased the formation of lung metastases in *BALB/c* nude mice [123]. Based on this, they further investigated that HIC1 overexpression inactivated the AKT pathway, whereas LCN2 overexpression induced AKT in MDA-MB-231 and MDA-MB-468 cells, thereby increasing their invasive potential. In comparison to other breast cancer subtypes, HIC1 is frequently epigenetically silenced in TNBC [123]. The authors proposed that this leads to increased invasiveness and may be linked to the aggressive nature of TNBC.

Clinically, low HIC1 expression in TNBC patients with a basal-like subtype correlates with poor overall survival and distant metastasis-free survival [123]. These findings indicate that silencing of HIC1 may be associated with subtype-specific prognoses of breast cancer by promoting LCN2 overexpression.

In a cohort of breast cancer patients, the LCN2 promoter was found to be unmethylated in 67% of cases. Among patients with TNBC, unmethylation of the promoter was detected in 9 out of 11 cases [124]. The authors found a correlation between unmethylated LCN2 promoters and significantly higher microvessel density, indicating increased tumor angiogenesis (Figure 2D). Additionally, these tumors displayed higher cell proliferation measured by Ki-67 staining, suggesting a more aggressive phenotype. The study proposes that LCN2 promoter unmethylation leads to elevated LCN2 expression, creating an epigenetic axis through which LCN2 regulation is altered in aggressive subtypes. Therefore, the methylation status of the LCN2 promoter may serve as a potential prognostic tool for identifying aggressive tumors and guiding therapeutic decisions.

A recently published study identified an epitranscriptomic mechanism regulating LCN2 expression in TNBC [125]. Wilms’ tumor 1-associated protein (WTAP), NUPR1 and LCN2 were found to be significantly upregulated in TNBC tissues, correlating with poor clinical outcomes. Silencing NUPR1 induced ferroptosis and simultaneously inhibited cell proliferation, migration and invasion. LCN2 was positively regulated by NUPR1, and its knockdown similarly promoted ferroptosis and suppressed malignant behavior. Mechanistically, WTAP enhanced NUPR1 expression through m^6^A RNA modification, stabilizing its mRNA stability via m^6^A reader protein eukaryotic translation initiation factor 3 subunit A (eIF3A) (Figure 2E). Collectively, the WTAP-NUPR1-LCN2 pathway suppresses ferroptosis and contributes to TNBC progression. This finding is of particular significance in TNBC as it uncovers previously unrecognized epitranscriptomic regulation. Given the therapeutic options for TNBC, targeting components of this pathway may provide a novel strategy to sensitize tumors to ferroptosis-inducing agents.

## 5. Organ-Specific Role of LCN2 in Metastasis

Several studies have shown that LCN2 has opposing effects on cancer progression depending on the type of cancer. It acts as an oncogene in breast [19], ovarian [129], gastric and oral squamous cell carcinomas [130], whereas it has tumor suppressor functions in colorectal [131] and pancreatic cancers [132]. The most common sites of TNBC metastasis are lung, brain, bone and liver [54,133,134]. It has been observed that metastases develop in different distant organs depending on the type of BC [135]. This so-called organ tropism is driven by the defined molecular and cellular environment of each organ rather than by spontaneous metastasis [136]. Despite great progress in recent decades, the complex reality of BC metastasis remains unexplained. Here, we focused on the precise role of LCN2 in the metastasis of TNBC to distant organs, as illustrated in Figure 3.

### 5.1. Lung

The lung is one of the most common sites of metastasis in TNBC patients, in contrast to non-TNBC patients [119]. An early study by Shi and colleagues showed that overexpression of LCN2 in murine 4T1 cells increased lung metastasis formation in vivo. Further in vitro experiments showed that LCN2 promoted migration and invasion via the PI3K/AKT pathway (phosphoinositide 3-kinase) [137]. However, Berger et al. showed that loss of *Lcn2* in MMTV-PyMT mice reduced primary tumor formation but not lung metastasis [141]. Leng et al. investigated the role of LCN2 in HER2^+^ BC cells and found that impaired LCN2 expression reduced invasiveness and migration of cancer cells. Intravenous injection of an anti-LCN2 antibody into breast tumor-bearing mice resulted in significant inhibition of lung metastasis, supporting the findings of Shi et al. [138]. These findings highlight the potential role of LCN2 in promoting BC cell invasion and lung metastasis, particularly in aggressive subtypes like TNBC. Understanding the context-dependent effects of LCN2 may offer novel therapeutic opportunities to inhibit metastatic spread.

### 5.2. Brain

TNBC has a strong tendency to spread to the brain, making it one of the most common sites of metastasis for this aggressive subtype. While HER2^+^ luminal breast cancer can also metastasize to the brain, this occurs less frequently [142]. A study by Chi et al. analyzed cancer cells in the cerebrospinal fluid (CSF) of patients with leptomeningeal metastasis (LM) originating from BC using single-cell RNA sequencing (scRNAseq) [139]. The leptomeningeal space is isolated from the blood circulation by the blood-CSF barrier and exhibits hypoxic conditions coupled with the delivery of micronutrients, ions and trace elements [143]. The scRNAseq analysis showed increased expression of genes associated with iron binding and transport in cancer cells, whereas immune cells express canonical iron transporter transcripts. Of these, LCN2 and solute carrier family 22 member 17 (SLC22A17) were exclusively expressed in cancer cells and not in macrophages. Functionally, they showed that LCN2 expression was induced by pro-inflammatory cytokines and promoted cancer cell growth in a mouse model of LM. In addition, cancer cells used LCN2/SLC22A17 to accumulate extracellular iron in the CSF and thus outcompete other cells in the leptomeninges. Iron chelation therapy was considered an appropriate therapy for LM, as it showed a beneficial survival effect in their in vivo model.

Another aspect of the influence of LCN2 in brain metastasis is highlighted by Liu et al. [140]. They elucidated how LCN2 secreted by neutrophils promotes tumor progression under the influence of c-Met [140]. They observed that c-Met signaling was activated in metastatic cancer cells, which then enhanced the expression of G-CSF, granulocyte-macrophage-colony stimulating factor (GM-CSF), CXC motif chemokine ligand 1 (CXCL1) and CXCL2. G-CSF and GM-CSF are regulators of neutrophil differentiation and promote their maturation [144]. Neutrophils stimulated by these factors exhibited increased LCN2 secretion, which subsequently contributed to tumor stemness [140]. Recent studies have shown that LCN2 increased cancer cell stemness through increased iron uptake [145].

Other studies have shown that immature and mature neutrophils can have suppressive (N1 neutrophils) or supportive (N2 neutrophils) functions that play an important role in tumor progression and metastasis formation [146]. In addition, tumor cells can recruit neutrophils into the microenvironment by expressing CXCL1 and CXCL2 [144]. The secretion of c-Met-dependent cytokines by brain metastatic cells resulted in the N2 phenotype of neutrophils [140]. Cancer stem cells are believed to possess a cellular plasticity to adapt to changes in the tumor microenvironment and a high capacity for self-renewal and tumor initiation [147].

### 5.3. Bone and Liver

Approximately 20–40% of patients with TNBC develop metastases from the primary tumor to the liver. When combined with metastases to other visceral organs, the frequency increases to 30–50% depending on the study, population and specific characteristics of the patients [54,142,148]. Additionally, around 19% of 1330 TNBC patients had bone-only metastases [149]. Interestingly, TNBCs have a lower incidence of bone metastasis (15.8%) compared to non-TNBCs [150].

To date, there are no substantial TNBC-specific studies investigating the role of LCN2 in liver or bone metastasis. Most research has focused on LCN2 in lung, brain and general metastatic phenotypes. This represents a significant gap, as the liver and bone are among the most clinically relevant metastatic sites in TNBC, strongly impacting patient prognosis and therapeutic strategies. Understanding whether LCN2 contributes to the colonization and progression of metastases in these organs could reveal novel mechanisms and potential therapeutic targets limiting disease spread.

## 6. Conclusions

LCN2 has emerged as a pivotal regulator in BC and its multifaceted influence on tumor progression, metastasis and cellular plasticity highlights its particular relevance for TNBC. Although many studies report its association with aggressive tumor behavior, the mechanistic understanding of how LCN2 contributes to TNBC progression, metastasis and therapy resistance is still limited. The functions of LCN2 appear to be highly tumor- and tissue-specific. It has been shown to promote invasion, metastasis and EMT in various cancers (including TNBC), but in some cancer types it appears to have neutral or even suppressive effects. These discrepancies are likely due to differences in tumor subtype, tissue origin and microenvironmental cues, making it difficult to predict whether findings from other BC subtypes or cancers apply directly. The role of LCN2 in TNBC has primarily been studied in vitro using MDA-MB-231 cells, which are the most commonly used TNBC cell line. According to Lehmann et al. MDA-MB-231 cells are classified as a mesenchymal stem-like subtype [41,47]. However, cell lines representing other molecular TNBC subtypes are seldom utilized in cancer research. Consequently, additional studies are required to examine the role of LCN2 across various TNBC subtypes in order to determine if there are subtype-specific differences and to gain a better understanding of how molecular heterogeneity influences its function.

Most studies focus on LCN2 expression at a single point in time, often in primary tumors or established cell lines. However, in TNBC metastasis, changes in basal LCN2 levels during interactions with the TME or with other cancer cells might be more important than its absolute expression level. It is still unknown whether LCN2 upregulation or downregulation occurs at specific metastatic stages, such as during EMT-driven invasion or MET-driven colonization. Future studies should examine the dynamics of LCN2 expression throughout the entire metastatic cascade in vivo.

Although LCN2 has been linked to increased vimentin expression, enhanced migration and invasive morphology in TNBC, the exact molecular pathways, whether through MMP-9 activation, iron homeostasis, or signaling pathway modulation, remain poorly defined. This lack of mechanistic clarity limits our ability to target LCN2-driven processes therapeutically. Previous studies have identified an association between LCN2 expression, chemoresistance and ferroptosis resistance in TNBC. The authors proposed that iron-chelation therapy or targeting the WTAP-NUPR1-LCN2 axis could help overcome these resistance mechanisms. However, current research remains largely confined to in vitro models. Future investigations should aim to evaluate the therapeutic potential of targeting the LCN2 pathway in combination with chemotherapeutic, immunotherapeutic, or iron-chelation approaches in vivo. In accordance with the 3R principle (replacement, reduction, refinement), patient-derived TNBC organoids represent a promising ex vivo model that faithfully recapitulates key features of the primary tumor [151]. This model could serve as a platform for investigating novel treatment approaches and assessing therapeutic efficacy.

Multiple putative LCN2 receptors add further complexity, raising the possibility of receptor-specific targeting if subtype-specific expression patterns can be identified; however, it is not known which receptor contributes to LCN2-mediated tumor progression, invasion, or metastasis. Additionally, it remains unclear whether TNBC cells secrete LCN2 and stimulate themselves in an autocrine manner, or if LCN2 is secreted by cells in the TME and acts as a paracrine stimulator. To fully characterize the downstream signaling pathways of LCN2, future investigations should examine the expression and distribution of its putative receptors in both cell line models and patient-derived samples. Such analyses would provide deeper insights into the mechanisms by which LCN2 exerts its effects on cancer cells and other cell types within the TME.

Different studies use diverse TNBC cell lines (e.g., MDA-MB-231 and BT-549), animal models and patient cohorts often without considering genetic and phenotypic differences between them. Consequently, findings on LCN2’s function and regulation in TNBC can vary widely, making cross-study comparison difficult and sometimes contradictory. While elevated LCN2 levels in serum or tumor tissue have been proposed as diagnostic or prognostic biomarkers in BC, there are no standardized thresholds, detection methods or prospective clinical trials validating its utility in TNBC. Similarly, although LCN2 is considered a potential therapeutic target, no targeted interventions against LCN2 are in clinical development for TNBC yet. Future research should aim to establish standardized experimental approaches and employ well-characterized TNBC models representing distinct molecular subtypes. The use of patient-derived organoids and xenograft models could provide more physiologically relevant insights into LCN2 signaling and function. Further studies are warranted to elucidate downstream pathways, receptor distribution, and the contribution of LCN2 to treatment resistance and TME interactions. Systematic validation of LCN2 as a biomarker and preclinical assessment of LCN2-targeted or combinatorial therapeutic strategies will be essential to determine its translational potential in TNBC management.

Despite increasing evidence supporting the involvement of LCN2 in metastatic progression, its role in organ-specific metastasis remains incompletely understood. Current studies have primarily linked LCN2 to enhanced metastatic dissemination to the lung and brain, suggesting a possible contribution to the establishment or maintenance of a pro-metastatic microenvironment in these tissues. However, the role of LCN2 in metastasis to other clinically relevant sites, such as the liver and bone, remains largely unexplored, despite the high prevalence and clinical impact of metastases in these organs among TNBC patients. Addressing this gap is crucial for advancing our understanding of LCN2-driven metastatic organ-tropism. Future investigation employing well-designed in vivo and ex vivo models, including organotypic culture systems, will be essential to elucidate whether LCN2 actively mediates site-specific metastatic colonization, or if its role varies depending on microenvironmental context. Such studies could provide valuable insight into the potential of LCN2 as a biomarker for metastatic risk stratification and as a target for therapeutic strategies aimed at preventing or limiting organ-specific disease spread in TNBC.

## Figures and Tables

**Figure 1 ijms-26-10938-f001:**
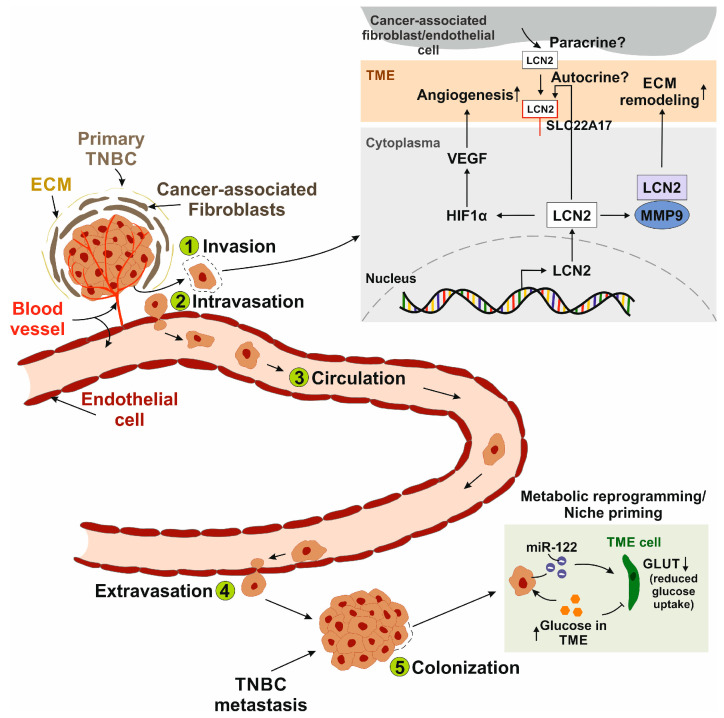
Schematic representation of the metastatic cascade and the role of LCN2 in each step. (1) Invasion: Cancer cells disseminate from the primary tumor, which is embedded in the extracellular matrix (ECM) and surrounded by cells of the tumor microenvironment (TME), such as cancer-associated fibroblasts. LCN2 interacts with matrix metalloproteinase 9 (MMP-9), leading to ECM remodeling. It remains unclear whether secreted LCN2 stimulates cancer cells in an autocrine manner via the putative LCN2 receptor solute carrier family 22 member 17 (SLC22A17) and/or acts in a paracrine manner when secreted by TME cells. (2) Intravasation: LCN2 promotes vascular endothelial growth factor (VEGF) expression via hypoxia-inducible factor 1-alpha (HIF1α), enhancing angiogenesis and vascular permeability, thereby facilitating the entry of metastatic cancer cells into the bloodstream. (3) Circulation: Circulating cancer cells adhere to the microvasculature by binding to endothelial cells when adhesive forces exceed the shear stress of blood flow. (4) Extravasation: Cancer cells attach to the luminal surface of the endothelium and migrate across it into the surrounding tissue. (5) Colonization: Metastatic cells establish secondary tumors at distant sites. A common strategy involves metabolic reprogramming or niche priming of cancer cells, for example, through the secretion of microRNA-122 (miR-122), which reduces glucose transporter (GLUT) expression in TME cells. As a result, glucose accumulates in the TME and becomes more available to tumor cells. LCN2 has been implicated.

**Figure 2 ijms-26-10938-f002:**
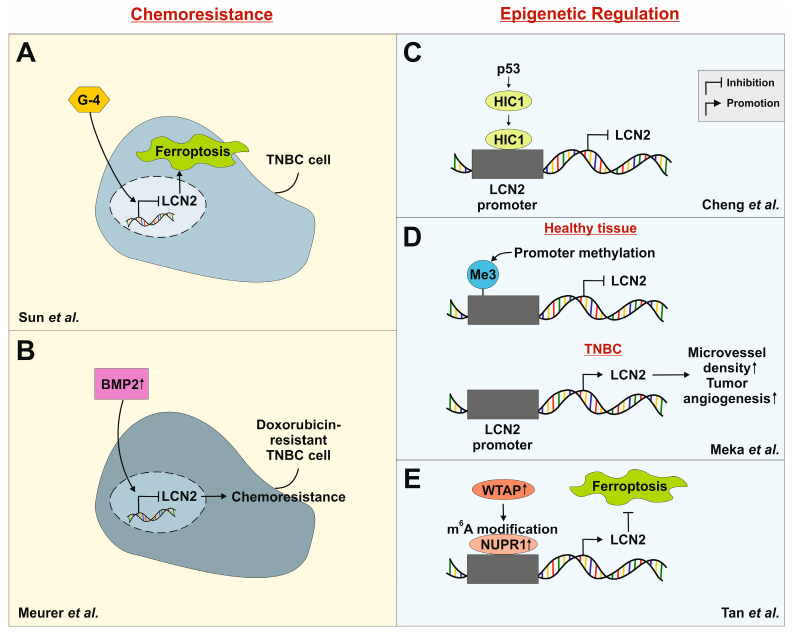
Schematic representation of the influence of LCN2 in chemoresistance and its epigenetic regulation. (**A**) Treatment with G-4, a derivative of roscovitine (a cyclin-dependent kinase inhibitor), reduces cell viability, proliferation, and migration of TNBC cells in vitro and in vivo. LCN2 expression is downregulated in treated cells, leading to ferroptosis [121]. (**B**) LCN2 is downregulated in doxorubicin-resistant TNBC cells, accompanied by increased bone morphogenetic protein 2 (BMP2) expression. BMP2 suppresses LCN2 expression. Thus, LCN2 downregulation appears to be a hallmark of acquired chemoresistance [122]. (**C**) LCN2 is regulated downstream of hypermethylated in cancer (HIC1). HIC1 is transcriptionally controlled by p53 and is frequently silenced in TNBC, contributing to increased invasiveness [123]. (**D**) Promoter methylation is an important mechanism of gene regulation. The LCN2 promoter is unmethylated in 9 out of 10 TNBC patient samples, leading to increased LCN2 expression and resulting in higher microvessel density and tumor angiogenesis [124]. (**E**) LCN2 is positively regulated via the WT1-associating protein (WTAP)/NUPR1 axis. WTAP enhances nuclear protein 1 (NUPR1) through m6A modification, and binding of this complex to the LCN2 promoter increases LCN2 expression, thereby suppressing ferroptosis [125].

**Figure 3 ijms-26-10938-f003:**
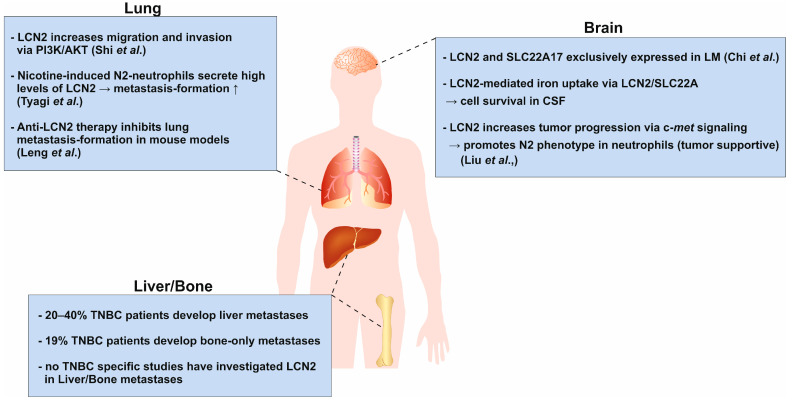
Schematic representation of the context-dependent role of LCN2 in organ-specific metastasis of TNBC. LCN2 exhibits opposing functions in cancer, acting as an oncogene in breast, ovarian, gastric, and oral squamous cell carcinomas, while displaying tumor-suppressive effects in colorectal and pancreatic cancers. TNBC most frequently metastasizes to the lung, brain, bone, and liver, a process influenced by organ-specific microenvironments (organ tropism). In the lung, overexpression of LCN2 promotes migration and invasion via PI3K/AKT signaling and enhances lung metastasis in vivo [137]. Antibody-mediated inhibition of LCN2 reduces metastatic burden, supporting its pro-metastatic role [138]. In the brain, LCN2 and its receptor SLC22A17 are highly expressed in cancer cells within the leptomeningeal space [139]. LCN2 supports cancer cell growth by facilitating iron acquisition and is induced by inflammatory cytokines. Targeting iron metabolism improves survival in vivo. Additionally, c-Met signaling in metastatic cells recruits and activates neutrophils via granulocyte-colony stimulating factor (G-CSF), granulocyte-macrophage-colony stimulating factor (GM-CSF), and C-X-C motif chemokine ligand 1 (CXCL1) and C-X-C motif chemokine ligand 2 (CXCL2), leading to increased LCN2 secretion, cancer stemness, and brain metastatic progression [140]. Regarding bone and liver metastasis, the role of LCN2 remains largely unexplored despite being clinically relevant metastatic sites in TNBC. This represents a major research gap with potential therapeutic implications. Understanding the context-dependent and organ-specific functions of LCN2 may uncover novel mechanisms driving TNBC metastasis and inform targeted interventions.

**Table 1 ijms-26-10938-t001:** Triple-negative breast cancer subtypes according to Lehmann et al. [40,41].

TNBC Subtype [40]	TNBC Subtype [41]	Characteristics	Cell Line	LCN2 Expression (nTPM) *
Basal-like 1	BL-1	↑ Ki67 [42] ↑ *MYC* activity (associated with worse survival) [43] ↑ *RhoA* activity (linked to poor prognosis) [43] ↑ basal cytokeratins (CK5/6, CK14, CK17) [44] Best overall survival among other TNBC subtypes [41]	HCC38	9.9
			HCC1143	284.2
			HCC1599	1225.9
			HCC1937	1796.1
			HCC2157	195.4
			MDA-MB-468	351.4
Basal-like 2	BL-2	Basal-myoepithelial phenotype [42] ↑ Growth-factor signaling (EGFR, Wnt/β-catenin) [40] ↑ Glycolysis and gluconeogenesis ↑ E2F2 pathway (associated with poor overall survival) [45] ↑ TGF-β pathway (associated with worse overall and disease-free survival) [46] ↑ Basal cytokeratins (CK5/6, CK14, CK17) [44]	HCC70	270.5
Mesenchymal	M	Genetic patterns responsible for cell motility and cell differentiation (Wnt, ALK pathway) [42] ↑ Extracellular matrix-receptor interactions [47] ↑ Genes involved in epithelial–mesenchymal transition [48] ↑ Resistance to chemotherapeutic agents [49]	BT-549	3.7
			Hs 578T	0.4
Mesenchymal stem-like		Genetic profiles associated with growth factor signaling pathways (*EGFR*, *PDGF*) [43] ↓ Proliferation-associated genes [47] ↑ Stem-cell-associated genes [47] ↓ Claudins [42] Pro-angiogenic gene expression (*VEGFC*, *SEMA3G*, *SEMA5A*) [50]	MDA-MB-231	16.4
			MDA-MB-436	289.0
			MDA-MB-157	0.3
Immunomodulatory		Gene ontologies related to immune cell processes (e.g., immune signal transduction) [42] ↑ TILs [51] ↑ IFN-α and IFN-γ pathways [43] Treatable with immune checkpoint inhibitors [51] More favorable prognosis compared to other subtypes [51]	DU4475	0.3
			HCC1187	16.4
			HCC1806	137.6
LAR	LAR	↑ Androgen receptor ↑ Hormonal signaling pathways (steroid synthesis, androgen/estrogen metabolism) [48] ↓ Ki67 [52] ↑ Luminal cytokeratins (CK7/8, CK18, CK19) [53]	MDA-MB-453	0.9

Abbreviations used: ↑ upregulated, ↓ downregulated, ALK = anaplastic lymphoma kinase, CK = cytokeratin, E2F2 = E2F transcription factor 2, EGFR = epidermal growth factor receptor, IFN-α/γ = interferon alpha/gamma, nTPM= normalized transcripts per million, PDGF = platelet-derived growth factor, RhoA = ras homolog family member A, SEMA = semaphorin, TGF-β = transforming growth factor beta, TIL = tumor infiltrating lymphocytes, VEGF = vascular endothelial growth factor. * The Human Protein Atlas, Cell Line Atlas—RNA seq expression data for breast cancer cell lines. Available at: https://www.proteinatlas.org. Last accessed 13 October 2025.

## Data Availability

No new data were created or analyzed in this study. Data sharing is not applicable to this article.

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
