# Peer review of "Lipocalin-2 in Triple-Negative Breast Cancer: A Review of Its Pathophysiological Role in the Metastatic Cascade"

_ijms, 2025, doi:10.3390/ijms262210938_

Round 1

Reviewer 1 Report

Comments and Suggestions for Authors

This is a well-conceived and timely review that effectively synthesizes the current understanding of LCN2's multifaceted role in TNBC metastasis. The authors have done an excellent job in structuring the content and identifying key knowledge gaps. Below are specific comments and suggestions to further strengthen the manuscript before publication.

The manuscript frequently references "Error! Reference source not found." for Table 1, Figure 1, and Figure 2. It is critical that the final version includes all these figures and the table. Their content, as described in the text, is essential for understanding the subtypes of TNBC and the proposed mechanisms of LCN2 action.

The conclusion section is present but could be more impactful. I suggest expanding it to provide a forward-looking perspective. Specifically, propose concrete future research directions based on the gaps identified (e.g., "Future studies should focus on... elucidating LCN2's role in liver/bone metastasis using TNBC-specific models," "investigating the dynamics of LCN2 expression throughout the metastatic cascade in vivo," or "clarifying the contributions of its different receptors").

The discussion on phenotypic plasticity (Section 3.2) is excellent. To enhance clarity, consider adding a sentence or two that explicitly distinguishes the concepts of EMT/MET from the maintenance of a "mesenchymal phenotype" and how LCN2 fits into these distinct but related processes.

    • P1, Abstract: "metastasic" should be "metastatic".

    • P1, Introduction: "it was firstly isolated" -> "it was first isolated".

    • P3: "well treatable" -> "responsive to treatment" or "treatable".

    • P5: "migration process is also known as extravasation" -> "this step, known as extravasation,...".

    • P7: "Migration cues required for tumor cell invasion through the stroma are provided by, among others, extracellular matrix (ECM) proteins..." -> "Migration cues for tumor cell invasion through the stroma are provided by various factors, including extracellular matrix (ECM) proteins...".

    • Please perform a thorough line-by-line edit for similar minor grammatical improvements.

Ensure all acronyms are defined upon first use. For example, "LM" for Leptomeningeal Metastasis on Page 12 is defined, but it's good practice to double-check all others.

Author Response

Dear Reviewer 1,

Please find attached our reponse to your thoughtful comments.

Regards

Ralf Weiskirchen

Reviewer 2 Report

Comments and Suggestions for Authors

The submitted article entitled “Lipocalin-2 in Triple-Negative Breast Cancer: A Review of Its Pathophysiological Role in the Metastasic Cascade” by, et al. document the role of Lipocalin-2 in TNBC highlighting its pathological and physiological influence on metastatic cascade. It provides a vivid as well as detailed account of the pathological, physiological role of LCN2, followed by its role in general breast cancer and then narrowing down to TNBC metastasis. The review provides a clear overview of the topic with well-defined illustrations. The author arranged the review article in a well-organized form, and the novelty lies in highlighting TNBC metastatic cascade. This review article can be considered for publication by modifying some minor reviews.

  1. The title should be “Lipocalin-2 in Triple-Negative Breast Cancer: A Review of Its Pathophysiological Role in the Metastatic Cascade” and not Metastasic.

  1. In the review article the authors gave information about the expression of LCN2 and innate immune cells plasticity. A separate paragraph on LCN2 and its role in modulation of immune response in context of TNBC (both adaptive and innate immune response) would add more lucidity for the reader.

Author Response

Dear Reviewer 2,

Please find attached our reponse to your thoughtful comments.

Regards

Ralf Weiskirchen

Reviewer 3 Report

Comments and Suggestions for Authors

As a subtype of breast cancer characterized by poor prognosis and limited treatment options, triple-negative breast cancer (TNBC) represents a significant clinical challenge. This review systematically examines the pathophysiological role of lipocalin-2 (LCN2) in TNBC, with a focus on its involvement in organ-specific metastasis, and highlights its potential for clinical translation, demonstrating substantial innovation in this research area. 

1. There is a spelling error in the title; it should be "Metastatic cascade" rather than "Metastasic cascade".

2. A study on LCN2 and TNBC in 2025 shows that aberrant iron metabolism mediated by LCN2 and TFRC modulates ferroptosis sensitivity in TNBC (Breast Cancer Res. 2025 Sep 26;27(1):165). Please carefully review the research progress over the past five years to help ensure all relevant citations are included.

3. The conclusion of the review appears somewhat abrupt. It is recommended to address the unresolved issues highlighted in the text concerning mechanistic research, clinical translation pathways, and organ-specific metastasis. On this basis, a more comprehensive discussion should be provided on the potential of LCN2 as a diagnostic biomarker and therapeutic target in TNBC, along with a forward-looking perspective on its prospects for future research and clinical application.

4. Please verify and correct the seven instances of "Error! Reference source not found" as well as all associated content.

Author Response

Dear Reviewer 3,

Please find attached our reponse to your thoughtful comments.

Regards

Ralf Weiskirchen

Round 2

Reviewer 3 Report

Comments and Suggestions for Authors

The manuscript has been improved after the revision. However, there are still a few instances of "Error! Reference source not found" appearing in the revised manuscript. I request that the authors pay close attention to these details and thoroughly verify the document before submission.

Author Response

We would like to express our gratitude to the reviewer for carefully reviewing our revised manuscript and for bringing to our attention the remaining “Error! Reference source not found” placeholders that appeared in our PDF file.

We have conducted a thorough line-by-line scan of both the Word and PDF documents, which includes the main text, figure captions, and tables. All broken cross-references have been repaired by correctly linking them to the corresponding numbered items in our reference management system. To prevent any future occurrences, we have created the PDF from a clean source file, compiled it twice, and validated it.

There are no longer any instances of the “Error! Reference source not found” message in the manuscript.

We apologize for the oversight and appreciate the diligence of the reviewer, which has helped us enhance the quality of our submission.